# Methods for Durability Testing and Lifetime Estimation of Thermal Interface Materials in Batteries

**Ralf Stadler \* and Arno Maurer †**

Polytec PT GmbH Polymere Technologien, D-76307 Karlsbad, Germany; a.maurer@polytec-pt.de
\* Correspondence: r.stadler@polytec-pt.de; Tel.: +49-7243-604-4191
† Present address: NTB Interstaatliche Hochschule für Technik Buchs, CH-9471 Buchs, Switzerland.

**Abstract:** To ensure sufficient thermal performance within electric vehicle (EV) batteries, thermal interface materials (TIMs), such as pastes or adhesives, are widely used to fill thermally insulating voids between cells and cooling components. However, TIMs are composite materials that are subject to degradation over the battery's lifetime. Using TIMs for battery applications is a new and emerging topic, creating the need to rapidly acquire knowledge about appropriate lifetime testing and evaluation methods, in close collaboration with the battery manufacturers. This paper reviews suitable methods for durability testing as well as basic modeling approaches which allow for the transfer of laboratory results to the longtime behavior of interface materials during a vehicle's lifetime.

**Keywords:** batteries; thermal management; thermal interface material; reliability testing; accelerated aging; lifetime estimation

---

## 1. Introduction: Thermal Management in Electric Vehicle Batteries

Recently, there has been a strong and increasing demand for innovative manufacturing concepts for electric and hybrid vehicle batteries. The design of a battery system from lithium-ion cells presents special challenges to thermal management [1,2]. As the performance and durability of the cells depend strongly on the temperature of their environment, the thermal management system has to care for the efficient dissipation of heat loss, as well as for the heat supply in case the batteries are cold. In operation, heat is generated when the system is being discharged due to accelerating, but also when charged at the charging station or during recuperation of braking energy. To avoid hot spots and to slow the thermal response within the battery pack, a large thermal mass and good inter-cell thermal conductivity are advantageous.

Heat delivery and dissipation can be provided in various ways [3]. Liquid-cooled systems have heat exchangers joined to the cells where the cooling medium absorbs the heat and conveys it to an external chiller. The heat transfer can be accomplished, either directly from the cells into a cooled baseplate, or via active or passive cooling sheets (located in between the cells) that are in turn thermally connected to the baseplate. Both mechanical and thermal connection is usually done by mechanical joining (bolts, clamps) or various welding procedures. Alternatively, thermally conductive adhesives and thermal interface materials (TIMs) provide a novel but already proven bonding solution [4]. There are numerous options for applying thermal adhesives and pastes in battery assembly [5,6]. Prismatic (hard case) cells can be mechanically fixed onto the cooling baseplate using thermally conductive structural adhesives, as shown in Figure 1a. When thermally connecting cooling sheets in between prismatic or pouch cells, as shown in Figure 1b, dispensable adhesives and fillers will be advantageous over foils and pads, which will have to be manually processed. Assembling the resulting modules on the battery pack base-frame is also possible using thermal interface materials, as shown in Figure 1c.

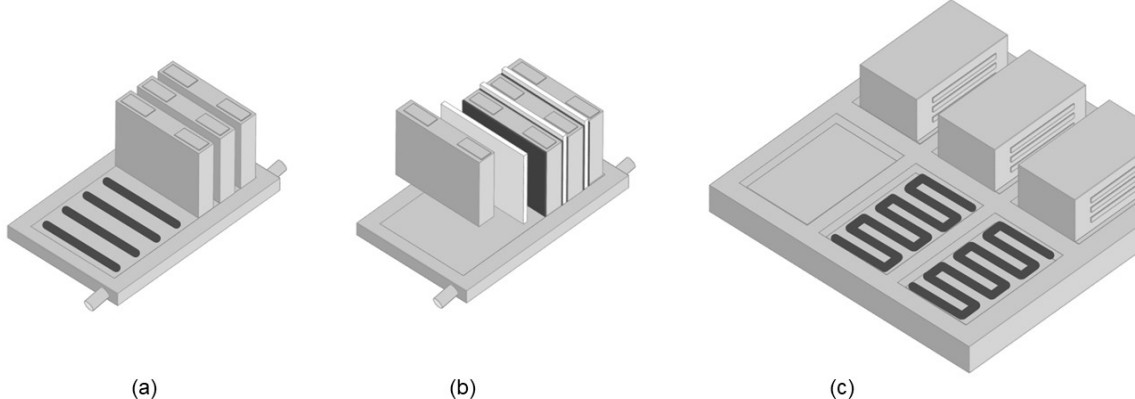

**Figure 1.** Thermally conductive bonding of prismatic cells onto cooled baseplate, (**a**) mounting of cooling sheets or plates between cells, (**b**) and assembly of modules on the base-frame (**c**) using thermal interface materials (black).

## 2. Thermal Interface Materials

### 2.1. Composition of TIMs

Thermal interface materials (TIMs) are composites made up of two or more components. Because the organic matrix, which is predominantly a polymer or liquid, generally has a low thermal conductivity of about 0.1 to 0.5 W/mK, it is complemented with thermally conductive fillers, such as aluminum oxide, aluminum nitride, graphite, metal particles, or similar materials. This type of composite thus combines the advantages of the polymers, such as low weight, good processing features, and corrosion resistance, with the thermal conductivity provided by the inorganic fillers. The resulting thermal conductivity of a composite material can reach between ca. 1 and 5 W/mK and is a function of the different thermal conductivities and the volume fractions of both matrix and filler. The basic functional and processing properties of these materials include thermal resistance, thermal conductivity, rheological behavior, density, and mechanical strength.

### 2.2. Types of TIMs

Commercially available types of TIMs are either non-curing thermal pastes and phase change materials (PCMs), ready-to-use thermal pads, or various polymer systems, such as adhesives, gels, and silicones that cure in place by a chemical reaction. Each of them has special advantages and drawbacks. Thermally conductive pastes are well-known from computer technology. They are easy to apply and to remove, and they feature a permanent thermal contact to the substrate surface due to their good intrinsic wetting properties. The same goes for PCMs, which can additionally absorb excess heat by melting. However, voids in battery modules can reach several millimeters of gap width. This requires thixotropic materials that are mechanically stable and non-sag, especially when considering dynamic loads like operational vibrations, shock impacts due to road holes while driving, and the varying inclination of vehicles when parked. Thermal pads, in contrast, are reasonably stable against mechanical loads, and they won't show any migration or separation during vehicle operation. However, they cannot be processed in an automated production line and they will not fit into gaps which exhibit high allowances. Thermally conductive adhesives and gels are also generally more resilient to aging than non-curing pastes but need a higher effort for metering and mixing before application. When cured, the parts are harder to disconnect in case for reworking or repairing.

## 3. Degradation of Thermal Interface Materials

A key risk factor in the development of thermal interface systems is the need to provide the material with sufficient thermal and mechanical stability to maintain its function, when used in a battery, during the vehicle's lifetime of 10–15 years. Thermal interface materials are exposed to various

operating conditions and environmental impacts during their service life. Both physical and chemical aging processes may occur, as shown in Figure 2.

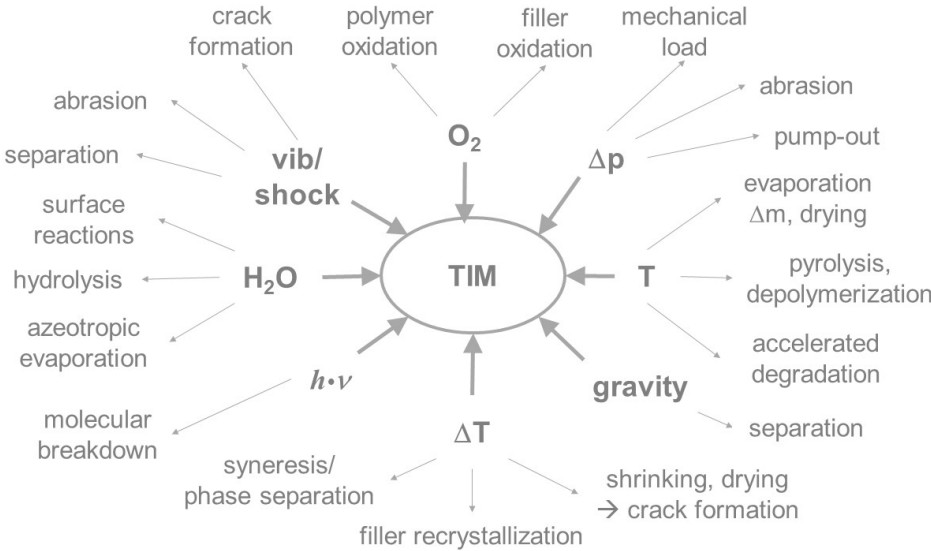

**Figure 2.** Chemical and physical factors responsible for aging and their impact on composite materials.

Polymer systems and composites generally have a finite lifetime under operating conditions. Chemical aging of a TIM will alter its properties, like its layer integrity, surface adhesion, elasticity, electrical insulation, thermal conductivity, and others. The alterations are due to changes in the molecular structure, the formation of new functional groups, chemical reactions between molecular components, or the breakdown of molecular bonds to form degradation products. All of these changes can be induced, or accelerated, by the absorption of energy in the form of heat. In addition, reactions with ambient media may occur. For example, organic compounds can react with oxygen from the air to form water and carbon dioxide; inorganic substances mostly form oxides. Physical, thermal, and thermomechanical loads may lead to the separation of the disperse and continuous phases [7], agglomeration or aggregation of particles, evaporation of the continuous phase, and macroscopic changes like migration or crack formation [8]. In particular, weakly or non-cross-linked materials tend to migrate, or separate, from fillers under vibrational stress and temperature changes. As a result, the thermal contact is interrupted under operating conditions. This can lead to the reduction, or in extreme cases, to the failure of heat transfer and therefore to a thermal overload within the battery. The following general failure mechanisms in TIMs are known or described in the literature [9]: migration [10], thermomechanical pump-out [11,12], delamination, cracking [13], decreasing coverage [14], pore formation, bleeding [15], dry-out [16], swelling or leaching by media, and oxidation [17].

In real-life operation, the influencing factors and aging mechanisms are complex. During vehicle use, the material is simultaneously subject to increased temperatures, temperature changes, shock/vibration, mechanical stress, and load changes as well as exposure to environmental media, such as atmospheric oxygen or air humidity. In the case of thermal pastes, which are classically used only for a layer thicknesses below 1 mm, the intended use in battery cooling and the resulting requirement for filling larger gaps up to several millimeters results in the additional risk of mechanical instabilities concerning inclination or vibration. Moreover, the TIM must be compatible with the common construction materials used in the automotive industry, like steel, aluminum, and various polymers and coatings, where no interactions or alterations of both the components and the TIM are allowed. Also, consideration must be given to aspects of electrochemical corrosion due to moisture loads and abrasion of the contact surfaces in reaction to vibration loads.

Unfortunately, the operational and environmental conditions within electric vehicle (EV) batteries have barely been investigated and described in the literature, and are mostly proprietary information of the manufacturers, although generally aging of batteries is of high research interest [18] and life cycle testing of electric vehicle battery modules has been standardized [19]. Despite ever-present operational vibrations and air moisture, batteries are subject to temperature changes resulting from both the environment and the heating/cooling during operation and charging. Figure 3 shows some temperature abundance distributions found in the literature regarding the typical operation of EV batteries within their lifetime, with most abundant temperatures typically between 10 °C and 50 °C.

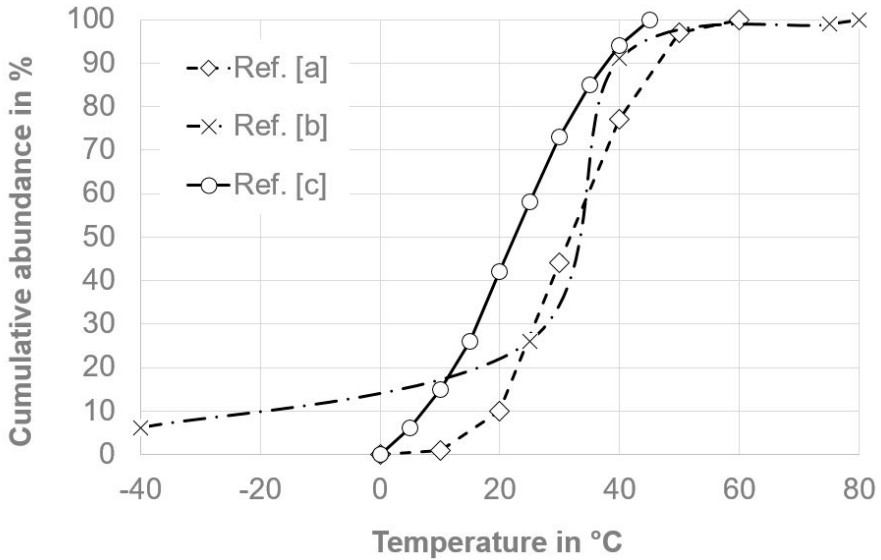

**Figure 3.** Typical cumulative temperature abundance within electric vehicle (EV) batteries over a lifetime. (**a**) Ref. [20]; (**b**) Ref. [21]; (**c**) Ref. [22].

## 4. Accelerated Ageing Test Methods

### 4.1. Overview

To advance the research and development of new materials, it is essential to simulate real-time aging processes with the aid of laboratory tests in a timely manner [23]. The aim of carrying out accelerated stress tests is to simulate the real loads in continuous operation as best as possible. This also requires new methods of long-term predictions with the help of lifetime models, which allow a transfer of laboratory test results collected within a short time window and a respective extrapolation over the entire operating period. The estimated conditions in a battery over its lifetime will determine the selection of the appropriate accelerated aging tests. Hence, test methods custom-tailored for the degradation phenomena occurring in TIMs over the lifetime of vehicle batteries are currently missing. Some approaches to investigation methods, which can be used to simulate aging processes and to characterize respective material changes, can be found in the literature, in international standards and corporate test specifications of semiconductor and automobile manufacturers. Common methods of accelerated aging include high-temperature storage (HTS), temperature cycling or shock (TC), climatic storage, alternating climate test, power cycling (PC), and various mechanical tests like vibration or shock tests. The evaluation of these tests is based on the monitoring of parameters such as the visual appearance, thermal conductivity, thermal or electrical resistance, thermal mass loss, oil separation, and mechanical characteristics, or rheological properties, e.g., complex viscosity, yield point. A detailed overview of the methods used for TIMs in electronics applications is provided in a recent literature review [24].

### 4.2. High-Temperature Storage

In high-temperature storage, the thermal stress upon the material is simulated. By using high, constant temperatures, the aging of the material is accelerated, which in turn allows for a prediction of its lifetime, provided that there is uniform reaction kinetics. A respective industrial standard series is ISO 60216/DIN EN 60216 [25], which is designed to estimate thermal degradation of electrical insulating materials. Here the relative temperature index (RTI) represents the numerical value of the temperature corresponding to the time where a selected property reaches a predetermined limit. For the underlying aging approach, physico-chemical processes are postulated which allow the application of the Arrhenius equation for the calculation of the degradation rate (see Section 5).

Temperature storage can also be performed at very low temperatures. Polymers and composites like TIMs will get brittle when applying low temperatures and are subject to mechanical degradation and crack formation. As a result, the possible embrittlement effects can be quantified and predictions about the stability of the material can be derived.

### 4.3. Temperature Cycling

In contrast to the constant temperature load when performing high-temperature storage, a temperature profile is applied during the cycle tests [26]. Respective industrials standards are known from semiconductor testing [27] and can be applied to TIMs [28]. This procedure simulates changing environmental temperatures [29] as well as temperature variations due to charging and discharging processes in batteries. A distinction is made between slow and continuous temperature changes in contrast to rapid temperature shocks where the change between high and low temperature, e.g., +80 °C and −40 °C, takes place within a very short time after a defined holding time.

### 4.4. Climatic Storage/Alternating Climate Test

Some stress tests feature a combination of temperature and humidity (damp-heat) storage. Respective industry standards are known e.g., for testing photovoltaic modules [30] and for semiconductors [31]. Damp-heat testing has also been applied to TIMs used for thermal connection of sensors in battery cells [32]. This type of highly accelerated temperature and humidity stress test (HAST) puts extraordinary stress upon the components. Depending on the conditions, a failure can be brought about after a few days. There are also combinations of temperature cycling and humidity storage called alternating climate test [33,34]. An extremely accelerated aging, as with this load combination, is important for the efficient development of new material systems.

### 4.5. Power Cycling

The power cycling test is the most realistic but also the most elaborate stress test. Here, the stress is simulated not only by a temperature profile, but by applying operating conditions, such as the charging and discharging of a battery, or the periodic heat generation of a power semiconductor chip. In contrast to passive/external temperature changes, the temperature change is actively affected by electrical heat output, which is introduced directly into the experimental setup [35]. A high number of cycles are feasible using this test. As a result, the influence of the parameter changes occurring during operation, such as temperature, gap width, or air humidity, can be properly simulated. Thermal pastes, especially, undergo various structural changes and degradation [36]. Power cycling can be carried out even under difficult conditions such as extreme temperature, abrupt temperature change, or high relative humidity.

### 4.6. Vibration Testing

By vibration of the engine or interaction of the vehicle with the road in operation, almost all components of an automobile are constantly subject to vibration. This type of stress [37] can be

simulated in a vibration test bench by setting certain vibration frequencies. There are respective industrial standards for components in general [38] but especially those designed for EV batteries [39].

*4.7. Complex/Combined Testing*

As described above, the situation in real operation is more complex therefore a combination of tests may be appropriate. In [24], a sequence of climate storage (HAST), temperature change (TC), and high-temperature storage (HTS) are recommended for the long-term assessment of TIMs for electronics applications. However, it is also mentioned that the reliability of an interface material depends crucially on the test setup or the parameters of the intended application and therefore there are no generally valid tests. Also, the dimensions and level of testing play a crucial role in evaluating the results [40]. Where a certain material may survive in small test setups, when it comes to large samples, thermal expansion may lead to additional voiding or cracking. Gravity effects may play a role in samples that are inclined according to parking on steep terrain, which won't be detected in samples that are stored horizontally. Hence, it is necessary to find evaluation criteria to assess the scaling of degradation and aging effects from laboratory samples into real-life sized setups.

## 5. Modeling for Lifetime Estimation

*5.1. General Considerations and Examples*

Reliability tests under accelerating conditions are crucial for generating lifetime data in short time periods. Releasing a reliable product to the market depends on this concept. From an economic point of view, it is impossible to test a product under operating conditions throughout the entire promised operating period of several years. That is why excessive loads are applied (e.g., elevated temperature, temperature changes, voltage, or humidity), which lead to accelerated aging, and therefore to a faster failure of the component. An acceleration factor is then calculated for each accelerated aging process in comparison to the real conditions of use. An acceleration factor with the value of two means a double load and, therefore, a halving of the service life. The goal of all models, which will be described below, is to predict the time of failure using acceleration factors. For different failure mechanisms, a large variety of models is available to determine the acceleration factor [41].

To make lifetime predictions based on test results, it is necessary to downgrade to simplified models. The basic approach is to accelerate aging at temperatures well above the intended operating temperature, assuming that the degradation mechanism remains the same at different temperatures [42]. A common method to make lifetime predictions is relating measurements to a kinetic model based on the Arrhenius equation (see Section 5.2). Input values can be, for example, decomposition temperatures from thermogravimetric analysis (TGA) measurements at different heating rates. Such an approach has been described by Ozawa, Flynn, and Wall [43] and can be applied to thermal interface materials [44]. The regression line yields the factors of the Arrhenius equation, which in turn helps to estimate the thermal lifetime of a TIM as a function of the operating temperature, as shown in Figure 4. The graph, in Figure 4, which has been plotted from such TGA measurements using a commercial thermal gap filler, illustrates that the estimated lifetime depends strongly on the temperature and is greater than 15 years in the common battery operation range.

The correlation between lifetime and test time can also be established in the case of tests accelerated by both temperature and moisture (see Section 5.3, Lawson model) and in the case of alternating temperatures (see Section 5.4, Coffin-Manson model). A simulation or modeling of alternating mechanical loads is more difficult; there have only been initial attempts, e.g., characterizing loads and failure mechanisms using mechanical test benches [45,46]. Concerning the results of thermomechanical loads, i.e., conditions where thermal and mechanical impacts occur at the same time, there is no recognized model [24].

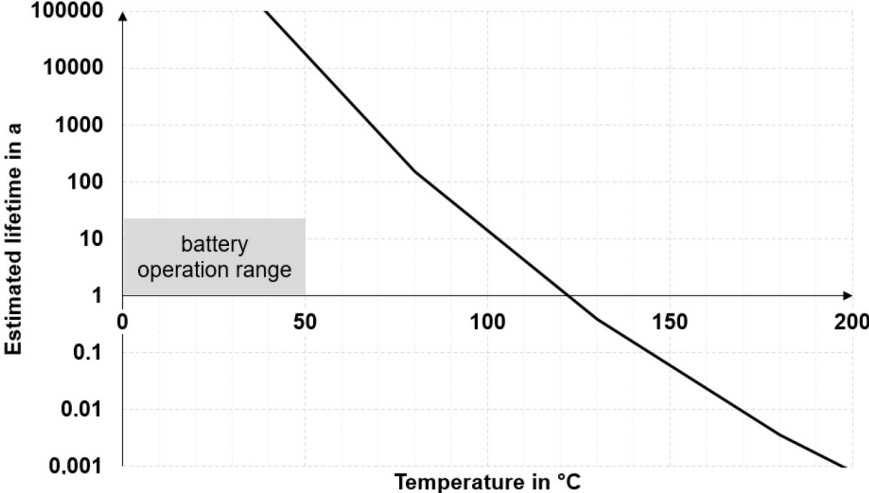

**Figure 4.** Modeling thermal lifetime of a battery gap filler vs. temperature [40].

*5.2. Arrhenius Model*

The Arrhenius equation [47] describes the quantitative relation between the reaction rate constant $k$ and the temperature $T$:

$$k = A \cdot e^{-\frac{E_A}{R \cdot T}} \tag{1}$$

The following simplifying assumptions are adopted in the equation:

- Monomolecular reaction
- Pre-exponential factor $A \neq f(T)$

Using the Arrhenius equation, the acceleration factor $A_T$ can be calculated from the activation energy $E_A$, the universal gas constant $R$ and the two temperatures $T_{test}$ and $T_{life}$:

$$A_T = e^{\left[\frac{E_A}{R} \cdot \left(\frac{1}{T_{test}} - \frac{1}{T_{life}}\right)\right]} \tag{2}$$

Here $T_{test}$ is the temperature applied during the aging test, and $T_{life}$ is the temperature during real operation. The acceleration factor $A_T$ corresponds to the ratio of the lifetime (operating time) $t_{life}$ and the test period $t_{test}$:

$$A_T = \frac{t_{life}}{t_{test}} \tag{3}$$

Assuming that different temperatures are occurring in different time periods during operation, this is called a temperature collective (see Figure 3). Using such an operating temperature collective, the overall acceleration factor of the temperature collective $A_{TK}$ can be described as the sum of the relative shares of all acceleration factors:

$$\frac{1}{A_{TK}} = \sum_i \frac{p_i}{A_{T,i}} \tag{4}$$

A simple exemplary calculation is feasible using this formula. Assuming a typical temperature collectively taken from [20], an operating time (lifetime) of 8000 h, an activation energy for the gap filler of 0.45 eV [21], and a test temperature of 85 °C, then the average acceleration factor is 8.4 which means that the lifetime is reached within 958 h of testing time, as shown in Table 1.

**Table 1.** Calculation of the necessary testing time based on accelerated aging.

| Temperature in °C | Abundance $p_i$ in % | Acceleration Factor $A_{T,i}$ | Abundance/Acceleration |
|---|---|---|---|
| 0 | 0% | 93.45 | - |
| 10 | 1% | 47.57 | 0.00 |
| 20 | 9% | 25.36 | 0.02 |
| 30 | 34% | 14.09 | 0.02 |
| 40 | 33% | 8.13 | 0.04 |
| 50 | 20% | 4.85 | 0.04 |
| 60 | 3% | 2.99 | 0.01 |
| Test temperature in °C | 85 | Lifetime in h | 8000 |
| Activation energy in eV | 0.45 | Test time in h | 958 |
| **Average acceleration factor** | | | **8.4** |

### 5.3. Lawson Model

Another model that takes into account both temperature and moisture is Lawson's model [47]. In addition, this requires the parameters $RH_{test}$ (relative humidity during accelerated aging), $RH_{life}$ (relative humidity in operation) and a material-specific constant $b$ to calculate the Lawson model acceleration factor $A_L$:

$$A_L = e^{[\frac{E_A}{R} \cdot (\frac{1}{T_{test}} - \frac{1}{T_{life}})] + b\ [(RH_{test})^2 - (RH_{life})^2]}$$ (5)

This model allows for estimating the influence of moisture on the aging process and the respective change in operating time to failure. The interaction of temperature and humidity clearly simulates the real conditions better and hence is valuable for the characterization of long-term properties of the TIMs.

### 5.4. Coffin-Manson Model

Unlike in the case of the Arrhenius and Lawson models, the acceleration factor $A_{CM}$ in the Coffin–Manson model [48] is calculated from the temperature differences occurring during temperature changes. The quotient of the temperature difference occurring during ageing, $\Delta T_{test}$, and the temperature difference occurring during operation, $\Delta T_{life,}$ is linked to the material constant $c$ as an exponent:

$$A_{CM} = \left(\frac{\Delta T_{test}}{\Delta T_{life}}\right)^c$$ (6)

Unlike the Arrhenius model, the Coffin–Manson model does not compare any specific temperature values but only relative temperature differences. The Coffin–Manson model is often used to describe failures caused by material fatigue in solid matter. However, this is related to information about crack formation and crack growth. Cracks in TIMs are undesirable but do not mean an immediate loss of function, as long as a sufficient coverage of the TIM and therefore thermal conduction is given. It seems that the Coffin–Manson model has not yet shown to be working for TIMs. Some kind of advanced modeling would involve combining the Coffin–Manson model with the impact of temperature and moisture in the manner of the Lawson model, thus generating a multi-dimensional design of experiments. Overall, practical models for lifetime prediction of interface materials under operating conditions still require a great deal of development.

### 6. Conclusions

The findings show that the overall lifetime prediction of TIMs used in EV batteries is complex. An iterative approach seems to be appropriate including the following steps. At first, key material properties of the TIMs under test are to be identified which would possibly deteriorate upon aging and therefore serve as failure criteria. Secondly, test procedures and respective boundary conditions

have to be selected which give a reasonable representation of the environment found in the battery, which can be suitably amplified to accomplish accelerated aging. Applying these procedures, the key properties identified before must be monitored to determine and quantify possible alterations. Finally, an appropriate lifetime calculation model must be selected that responds to the respective loads (temperature, humidity, pressure, etc.) that were applied during the test. When extrapolating and drawing general conclusions from the results, additional care must be taken regarding the sample dimensions (sample area and bond line thickness) as well as the testing level applied (from bulk material test to lab or application-scale to field testing level). In summary, to obtain significant knowledge and experience regarding long-term TIM durability, there is still a strong need for research.

**Author Contributions:** Both authors contributed equally to the literature research, evaluation, and discussion of the results.

**Funding:** This research was funded by the German Federal Ministry for Economic Affairs and Energy through the Central Innovation Program SME (Zentrales Innovationsprogramm Mittelstand—ZIM) under the grant no. ZF4344701 DE6.

**Conflicts of Interest:** The authors declare no conflict of interest.

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
