# Peer review of "Methods for Durability Testing and Lifetime Estimation of Thermal Interface Materials in Batteries"

_batteries, doi:10.3390/batteries5010034_

Round 1

Reviewer 1 Report

Dear author,

the paper is well written and gives a good overview on the current state of the art.

There are some minor errors to be corrected:

Figure 4: The reference is not given / Error Code on Reference

Line 239: "can also been established" needs to be corrected to "can also be established"

There are some points, which need clarification:

Line 243 - 244, 289-290: You use this sentence two times and it is not very clear what you mean here with combined "thermo-mechanical loads". What is "combined" in your terminology? Moreover the Coffin-Manson model you cite is for thermo-mechanical loads. Now you say that there is no model. Probably you mean, that the Coffin-Manson approach is not applicable to TIM materials? Or it is not yet shown to work for TIMs? Or it is shown not to work for TIMs? 

Line 266 and Table 1: You give a table on acceleration for the assumed activation energy of 0.45 eV: Why is this assumtion relevant? What is the use of displaying a result with this hypothetic value? If it is an exemplary demonstration of the approach, then you should name the table in this way. Or the relevance of the used activation energy should be explained.

Figure 4: It is not clear, where the acceleration factor for this figure comes from. Does the graph have any realistic meaning?

Author Response

Dear Reviewer,

thank you very much for your diligent comments and corrections which we were happy to implement in the manuscript.  

With regard to combined loads and the Coffin-Manson approach we have tried to unify, simplify and clarify the respective paragraphs. They read now as follows: 

"Much more difficult is a simulation or modeling of alternating mechanical loads. There are only first attempts, e.g. to characterize loads and failure mechanisms using mechanical test benches [45, 46]. With regard to the results of thermomechanical loads, i.e. conditions where thermal and mechanical impacts occur at the same time, there is also no recognized model [24]. 

"The Coffin-Manson model is often used to describe failures caused by material fatigue in solid matter. However, this is related to information about crack formation and crack growth. Cracks in TIMs are undesirable but don´t mean an immediate loss of function, as long as a sufficient coverage of the TIM and thus thermal conduction is given. It seems that the Coffin-Manson model has not yet shown to be working for TIMs."

Also, we have improved the description to table 1 (these calculations are in fact exemplary, the activation Energy is given in the reference) 

"A simple exemplary calculation is feasible using this formula. Assuming a typical temperature collective taken from Ref. [20], an operating time (life time) of 8,000 h, an activation energy for the gap filler of 0.45 eV [21], and a test temperature of 85 °C, …"

and Fig. 4 (this is based on real measurements) als follows

"...which in turn helps estimating the thermal lifetime of a TIM as a function of the operating temperature (Fig. 4). The diagram, which has been plotted from such TGA measurements using a commercial thermal gap filler, illustrates that the estimated lifetime depends strongly on the temperature and is greater than 15 years in the common temperature range where the battery is operated."

We hope this will help to clarify the manuscript, 

Thank you and Best Regards

Arno Maurer and Ralf Stadler

Reviewer 2 Report

The possible influences on the ageing behaviour are described in great detail and possible applicable models for the kinetics have been developed. What I miss is an example for the experimental approach to the quantitative consideration of ageing behaviour (kinetics). Even if it is correctly written in the summary that the ageing behaviour is very complex and cannot be mapped by one model, it should at least be shown exemplarily how the ageing behavior can be modelled.

Author Response

Dear Reviewer,

thank you very much for your diligent comment and correction.

We have improved the description to table 1 (these calculations are in fact exemplary)

"A simple exemplary calculation is feasible using this formula. Assuming a typical temperature collective taken from Ref. [20], an operating time (life time) of 8,000 h, an activation energy for the gap filler of 0.45 eV [21], and a test temperature of 85 °C, …"

and Fig. 4 (this is based on real measurements) as follows

"...which in turn helps estimating the thermal lifetime of a TIM as a function of the operating temperature (Fig. 4). The diagram, which has been plotted from such TGA measurements using a commercial thermal gap filler, illustrates that the estimated lifetime depends strongly on the temperature and is greater than 15 years in the common temperature range where the battery is operated."

Regarding the mathematic approach used (Ozawa, Flynn, and Wall) used, we did not show the detailed calculation here. Yet to be honest, more complex models are not at reach at the moment but may be developed and published at a later point of time. 

We hope this will help to clarify the manuscript,

Best Regards

Arno Maurer and Ralf Stadler

Reviewer 3 Report

Some reference errors remain in text (fig.4)

Interesting overview paper. It would enhance the paper also to mention phase-change materials for thermal management.

Author Response

Dear Reviewer,

thank you very much for your diligent reading and comments.

We have added some remarks about PCMs in chapter 2 and corrected the missing reference. 

"Commercially available types of TIMs are either non-curing thermal pastes and phase change materials (PCMs), ready-to-use thermal pads, or various polymer systems like adhesives, gels and silicones that cure in place by a chemical reaction. Each of them has his special advantages and drawbacks. Thermally conductive pastes are well-known from computer technology. They are easy to apply and to remove, and they feature a permanent thermal contact to the substrate surface due to good intrinsic wetting properties. The same goes for PCMs which can additionally absorb excess heat by melting"

We hope this will help to clarify the manuscript,

Best Regards

Arno Maurer and Ralf Stadler